# Application of Peak Glucose Range and Diabetes Status in Mortality Risk Stratification in Critically Ill Patients with Sepsis

**DOI:** 10.3390/diagnostics11101798

**Published:** 2021-09-29

**Authors:** Kai-Yin Hung, Yi-Hsuan Tsai, Chiung-Yu Lin, Ya-Chun Chang, Yi-Hsi Wang, Meng-Chih Lin, Wen-Feng Fang

**Affiliations:** 1Division of Pulmonary and Critical Care Medicine, Department of Internal Medicine, Kaohsiung Chang Gung Memorial Hospital, Chang Gung University College of Medicine, Kaohsiung 83301, Taiwan; redrosahung@yahoo.com.tw (K.-Y.H.); yhtsai@cgmh.org.tw (Y.-H.T.); chiungyu@cgmh.org.tw (C.-Y.L.); y7817@cgmh.org.tw (Y.-C.C.); yihsi@cgmh.org.tw (Y.-H.W.); mengchih@cgmh.org.tw (M.-C.L.); 2Department of Nutritional Therapy, Kaohsiung Chang Gung Memorial Hospital, Kaohsiung 83301, Taiwan; 3Department of Respiratory Therapy, Kaohsiung Chang Gung Memorial Hospital, Chang Gung University College of Medicine, Kaohsiung 83301, Taiwan; 4Department of Respiratory Care, Chang Gung University of Science and Technology, Chiayi 61363, Taiwan

**Keywords:** diabetes mellitus, peak glucose level, sepsis

## Abstract

The effects of diabetes and glucose on the outcomes of patients with sepsis are somewhat conflicting. This retrospective study enrolled 1214 consecutive patients with sepsis, including a subpopulation of 148 patients with immune profiles. The septic patients were stratified according to their Diabetes mellitus (DM) status or peak glucose level (three-group tool; P1: ≤140 mg/dL, P2: 141–220 mg/dL, P3: >220 mg/dL) on day 1. Although the DM group had a lower hazard ratio (HR) for 90-day mortality compared to non-DM patients, the adjusted HRs were insignificant. The modified sequential organ failure assessment-glucose (mSOFA-g) score can predict 90-day survival in patients with and without diabetes (β = 1.098, *p* < 0.001; β = 1.202, *p* < 0.001). The goodness of fit of the mSOFA-g score was 5% higher than the SOFA score of the subgroup without diabetes. The SOFA score and human leukocyte antigen-D-related (HLA-DR) expression were comparable between the groups. The P3 group had lower HLA-DR expression on days 1 and 3 and a higher 90-day mortality. The three-group tool was useful for predicting 90-day mortality in patients with separate Kaplan-Meier survival curves and mortality HRs in the construction and validation cohorts. The peak glucose level, instead of diabetes status, can be used as an easy adjunctive tool for mortality risk stratification in critically ill septic patients.

## 1. Introduction

Sepsis is a life-threatening organ dysfunction resulting from a dysregulated host response to infection [1], and can lead to circulatory, cellular, and metabolic abnormalities [2]. Sepsis occurrence or severity can be evaluated on the basis of the sequential organ failure assessment (SOFA) score, systemic inflammatory response syndrome (SIRS) criteria, and quick sequential organ failure assessment (qSOFA) score [3]. However, these scores do not include glucose levels, which may also be abnormal during sepsis due to metabolic changes.

Diabetes mellitus is a common comorbidity with an increasing incidence [4] in patients with sepsis. Patients with DM are prone to sepsis due to an impaired innate immune response [5]. However, the effects of diabetes on the outcomes of patients with sepsis are somewhat conflicting [6,7,8]. Hyperglycemia in septic patients without diabetes is associated with increased mortality and the development of DM after an episode [9]. Hyperglycemia may have different clinical implications in patients with and without diabetes [10] and the optimal blood glucose range during critical illness may differ [11]. A lower blood glucose level in the presence of diabetes may sometimes lead to a worse outcome. Almost U-shaped curves were noted for outcomes according to admission glycemia among cardiac patients with diabetes [12]. The current interest in personalized targets in the presence of diabetes is also of interest for comparison [13,14]. Immune derangements in these diseases may play an important role [15].

Sepsis can induce stress hyperglycemia in both patients with and without DM [16]. Patients with sepsis who are administered intensive insulin treatment also have a high risk of hypoglycemia and hyperglycemia [17]. Peak glucose values may be associated with sepsis outcomes, but are seldom discussed. With sepsis progression or resolution, the peak glucose level on the first day of ICU admission may differ. We hypothesized that there is an association between peak glucose levels and mortality risk in critically ill septic patients with or without DM. The association with outcomes could be related to the clinical parameters or immune status. The purpose of our study is to investigate the application of the peak glucose range and diabetes status in mortality risk stratification in critically ill patients with sepsis. In addition, we proposed a new tool, the modified sequential organ failure assessment-glucose (mSOFA-g) score, to facilitate mortality risk stratification.

## 2. Materials and Methods

### 2.1. Setting

This retrospective study enrolled patients admitted to adult medical intensive care units (ICUs; total 34 beds) of Kaohsiung Chang Gung Memorial Hospital, a 2700-bed tertiary hospital in Southern Taiwan, between August 2013 and January 2017, as the construction cohort. Consecutive patients with sepsis admitted to the medical ICU between January and August 2020 were included in the validation cohort.

### 2.2. Study Design

This retrospective cohort study screened patients through a review of medical records. Consecutive septic patients who met the Third International Consensus Definitions for Sepsis and Septic Shock (Sepsis-3) criteria [1] and with available data on peak glucose levels on day 1 of ICU admission were enrolled as the construction cohort. A part of the sepsis cohort during the first study period also investigated the clinical factors, biomarkers, and immune responses that predicted outcomes, as shown in our previous studies [18,19,20,21,22]. The validation cohort consisted of consecutive patients with sepsis, regardless of the treatment strategy. The study was approved by the Institutional Review Board of Chang Gung Memorial Hospital. We confirmed that all methods were performed in accordance with the relevant guidelines and regulations.

### 2.3. Definitions and Criteria

The peak glucose level on day 1 was defined as the highest glucose value measured during the first day of the ICU stay. The values included fingerstick or plasma glucose levels measured before or after meals. The P1 group was defined as having a peak glucose level of up to 140 mg/dL. The P2 group had a peak glucose level of 141–220 mg/dL. The P3 group was defined as having a peak glucose level greater than 220 mg/dL. The three ranges of peak glucose values (≤140, 141–220, >220) are based on the references in the literature [23,24]. We defined the P3 group as having a peak glucose level above 220 mg/dL because this level exceeded the renal threshold and was the point at which glycosuria occurred [24].

Diabetes was defined as a diagnosis of DM, regardless of the treatment type before ICU admission. Patients without recognized diabetes were classified as having no previous DM or non-DM. The glycemic control method was at the discretion of the physician caring for the patient in the ICU. We did not routinely use a standardized protocol (e.g., intensive insulin therapy) for glucose control in the ICU. However, we usually targeted an upper blood glucose level ≤180 mg/dL, as suggested by the guidelines [25]. Intensive insulin therapy has not been suggested for glycemic control in critically ill patients with sepsis due to the risk of hypoglycemia [26,27] and is not recommended for widespread adoption in medical ICUs [28,29]. Registered dietitians routinely checked patients’ nutritional status and provided nutritional recommendations based on the patients’ medical conditions during their stay in the ICU. If there were no contraindications, the patients received enteral feeding as soon as possible. In this pragmatic study, we did not exclude patients receiving parenteral nutrition, steroid administration, and insulin therapy.

Mortality was defined as death in hospital or discharge under critical conditions. The comorbidities such as coronary artery disease, history of stroke, hypertension, chronic obstructive pulmonary disease (COPD), cancer, chronic kidney disease (CKD), liver cirrhosis, and diabetes mellitus were collected from medical records. The serial changes of severity scores were measured by systemic inflammatory response syndrome (SIRS), sequential organ failure assessment (SOFA), and quick sequential organ failure assessment (qSOFA) on day 1 and day 3. All the severity scores were documented in medical records.

### 2.4. Data Collection

Clinical data, including data on diabetes status, peak glucose level on the first day, SOFA score [3,30,31], sub-scores, acute physiology and chronic health evaluation II (APACHE II) score [32,33], Charlson comorbidity index (CCI), underlying comorbidities, and other clinical factors, were retrieved from medical records. The immune profiles of a subset (*n* = 148, all in the construction cohort) of the enrolled patients who had participated in our previous study of the immune dysfunction score [18] were also analysed.

### 2.5. Immune Status Measurement

Human leukocyte antigen-D-related (HLA-DR) monocyte expression was measured according to the protocol described in the Appendix A and in our previous studies [18,34].

### 2.6. Statistical Methods

Patient demographics, clinical characteristics, and outcomes were summarized as frequencies and percentages for categorical variables. Median and interquartile ranges (IQRs), or 25th and 75th percentiles, were provided if appropriate. Because continuous variables were most likely not normally distributed, the Mann-Whitney U or Kruskal-Wallis tests were used when appropriate. Differences between the DM and non-DM groups were analyzed using Student’s t-test or the Mann-Whitney U-test, as appropriate, for continuous variables and the chi-squared test for categorical variables. Comparisons among the three groups were performed using Pearson’s chi-square tests and one-way analysis of variance (ANOVA), as appropriate. Pairwise comparisons were performed using ANOVA, with adjustments for multiple comparisons using Tukey’s range test for post-hoc comparisons. Kruskal-Wallis tests were used as a non-parametric alternative to ANOVA for non-normally distributed continuous variables. Dunn–Bonferroni pairwise tests were performed for post-hoc comparisons.

To assess the survival outcomes between groups, Kaplan-Meier (K-M) survival curves were constructed, with log-rank tests used for comparisons between groups. Mortality hazard ratios between groups were also compared using Cox proportional-hazards regression analysis and crude hazard ratios and adjusted hazard ratios for 90-day mortality. Independent variables (used to calculate the adjusted HR) included the peak glucose on day 1, DM status, and three-group tool (adjusted for age, sex, body mass index (BMI), APACHE II, CCI, coronary artery disease, history of stroke, hypertension, chronic obstructive pulmonary disease (COPD), cancer, chronic kidney disease (CKD), and liver cirrhosis).

Finally, we created a modified sequential organ failure assessment (mSOFA-g) score that included the peak glucose level score. The SOFA score is made up of six sub-scores, each representing an organ system (respiratory, cardiovascular, hepatic, renal, coagulation, and neurological systems). The sub-SOFA score ranged from 0–4 points for each item and the total SOFA score was 0–24 points. We propose the m-SOFA-g score, which consists of the SOFA score and an additional organ system (metabolic, by means of peak glucose level score). The maximal peak glucose level score is four points, which in this research, was assigned to P3 patients because of their highest 90-day mortality. The peak glucose level score included the P1 group (peak glucose level ≤ 140 mg/dL), which equaled one, P2 group (peak glucose level > 140 and ≤ 220 mg/dL) which equaled two, and P3 group (peak glucose level > 220 mg/dL), which equaled four. The mSOFA-g scores were equal to the sum of the SOFA scores and the peak glucose level scores. Logic regression was used to analyse the odds ratios of mortality related to the SOFA and mSOFA-g scores. The Nagelkerke R-squared was used to explain the model fit of the variables. Statistical significance was set at a two-sided *p* value < 0.05. All data were analyzed using IBM SPSS Statistics for Windows (version 22.0; IBM Corp., Armonk, NY, USA).

## 3. Results

### 3.1. Baseline Characteristics and Mortality by Different Glucose Metrics

Consecutive patients admitted to the medical ICU during the first and second study periods were screened. After excluding patients without sepsis or with incomplete data (e.g., peak glucose level), 722 and 492 patients were analyzed in the construction and validation cohorts, respectively (Figure 1). Demographic data and baseline characteristics are shown in Table 1 (construction cohort) and Appendix A (validation cohort).

#### 3.1.1. DM vs. Non-DM

Compared to non-DM patients, patients with DM were older (comparable age in the validation cohort) and had a higher BMI. As shown in Table 1, the DM group had a higher incidence of coronary artery disease, hypertension, and chronic kidney disease. The non-DM group had a higher incidence of cancer and male patients (comparable male percentage in the validation cohort). Although the APACHE II score and CCI were higher (but a worse APACHE II score and CCI were found in the DM group in the validation cohort), the DM group had lower crude 7-day and 28-day mortality rates than the non-DM group in the construction cohort. The 90-day mortality rate in the construction cohort was comparable. In the validation cohort, there were comparable crude 7-day, 28-day, and 90-day mortalities between the DM and non-DM groups. Therefore, the 90-day mortality rate was comparable in septic patients regardless of the presence of DM.

#### 3.1.2. Grouping Based on the Peak Glucose Level Range (Three-Group Tool)

Assessment of the peak glucose levels on day 1 (Table 1) revealed differences among the groups in terms of age, BMI (comparable BMI in the validation cohort, Appendix A), APACHE II, and CCI. Patients in the P3 group had the highest baseline glucose and glycated hemoglobin (HbA1c) levels. The incidence of hypertension was higher in this group. The crude 90-day mortality rates were higher in the P3 group than in the P2 group in the construction cohort. There was a borderline higher crude 90-day mortality rate in the P3 group in the validation cohort.

### 3.2. Crude and Adjusted Hazard Ratios for Mortality Comparisons via Cox Proportional Hazards Regression

#### 3.2.1. DM vs. Non-DM

As shown in Table 2 for the construction cohort, crude mortality hazard ratios were lower in DM patients regarding 7-day, 28-day, and 90-day (Cox regression) mortality compared with that in non-DM patients. After adjusting for baseline characteristics (model 1), the difference persisted. However, after additional adjustment for comorbidities, the mortality hazard ratios were comparable regarding 28-day and 90-day mortality between DM and non-DM patients. In the validation cohort (Appendix A), the crude and adjusted mortality hazard ratios were comparable for with and without DM.

#### 3.2.2. Risk Factors for 90-Day Mortality

Cox regression analysis was performed for 90-day mortality. The crude and adjusted hazard ratios (Model 1: adjusted for age, sex, BMI, APACHE II, and CCI; Model 2: adjusted for age, sex, BMI, APACHE II, CCI, coronary artery disease, history of stroke, hypertension, COPD, cancer, CKD, and liver cirrhosis) are shown in Table 3 and Appendix A. The three-group tool can discriminate the 90-day mortality in the total population and in patients with or without DM (after adjustment for baseline risk factors). Increased sepsis severity scores and decreased HLA-DR expression were also associated with mortality. DM status was not significantly associated with mortality in the total population. The analysis of the validation cohort revealed similar results. The three-group tool can discriminate between the 90-day mortality in the total population and in patients with DM (after adjustment for baseline risk factors; Appendix A).

#### 3.2.3. mSOFA-g Score

Table 4 shows the prediction effects of the mSOFA-g and traditional SOFA scores in the construction and validation cohort groups. The mSOFA-g scores on day 1 still significantly predicted the 7-day, 28-day, and 90-day mortality (*p* < 0.001); however, it did not increase the r-squared by more than 5% compared to the traditional SOFA score. Analysis of the validation cohort revealed similar results. The mSOFA-g score on day 1 still significantly predicted the 7-day, 28-day, and 90-day mortality; however, it did not increase the r-squared by more than 5% compared to the traditional SOFA scores.

As shown in Table 5 for the construction cohort, both the mSOFA-g scores and SOFA scores on day 1 significantly predicted the 7-day, 28-day, and 90-day mortality of patients with or without diabetes. The r-squared of the mSOFA-g score on day 1 for patients without diabetes increased by more than 5% compared to the SOFA scores on day 1 (7-day: from 0.179 to 0.235, 5.6%; 28-day: from 0.141 to 0.203, 5.6%, 6.2%; 90-day: from 0.098 to 0.157, 5.9%). In patients with diabetes, the r-squared increased the most in terms of the 90-day mortality by 4.8% (from 0.047 to 0.095, 4.8%). In the validation group, both mSOFA-g and SOFA scores on day 1 significantly predicted the 7-day, 28-day, and 90-day mortality rates of patients without diabetes. In patients with diabetes, both the mSOFA-g and SOFA scores on day 1 significantly predicted only the 90-day mortality. The r-squared of the validation group did not increase above 5% for the mSOFA-g score compared to the SOFA score on day 1.

### 3.3. Differences in 90-Day Kaplan-Meier Survival Curves and Mortality Hazard Ratios Based on Different Glucose Metrics

#### 3.3.1. DM vs. Non-DM

Figure 2 shows the 90-day survival curves, in which patients with DM had a better crude 90-day survival (Figure 2A) in the construction cohort. As mentioned before, the adjusted HRs were comparable between groups. There were comparable survival curves between the groups in the validation cohort (Figure 2C).

#### 3.3.2. Grouping Based on the Peak Glucose Level Range (Three-Group Tool)

The Kaplan-Meier survival plot showed differences in the curves between the P2 and P3 groups (Figure 2B). Comparisons of the hazard ratios between the groups using Cox regression models confirmed this phenomenon. Compared to the P3 group, the mortality rate was lower in the P2 group. The K-M curves in the validation cohort were similar (Figure 2D).

## 4. Discussion

In this study, we present four key findings. First, septic patients with DM were not associated with higher HRs for 90-day mortality. We observed comparable 90-day survival rates in critically ill septic patients with and without DM in the adjusted analyses. Second, our proposed tools according to the peak glucose level range on day 1 (three-group tool) can be helpful for risk stratification. Third, we created a modified sequential organ failure assessment-glucose score that increased the goodness of fit above 5% in the subgroup of patients without diabetes. Fourth, different underlying presentations of sepsis severity score and host immune status were noted among the groups. Although whether any difference in mortality could be explained by differences in immune response remains speculative, our findings shed light on the field.

The 90-day survival curves showed that the patients with DM had better survival than patients without diabetes. However, after adjustment for age, sex, BMI, and comorbidities, the difference disappeared. This might be because the patients with diabetes had more comorbidities compared to patients without diabetes. In our previous study, we noted that the increase of HbA1c had a tendency to decrease the mortality in patients with sepsis and a poor nutrition status, and cancer was related with a significant independent increase in mortality risk [35]. In this study, patients with DM had higher HbA1C levels and fewer cancers than patients without DM. This might explain why patients with DM had better survival before comorbidities were adjusted.

Furthermore, we hypothesized that the peak glucose level on day 1 or within the first three days could reflect the progression or resolution of sepsis. We found that the tool when using peak glucose levels within the first three days also had promising results. However, it appeared that we needed to wait for three whole days to obtain the data on the peak blood glucose, to barely predict the prognosis of sepsis. Our study found that our three-group tool using the peak glucose levels on day 1 was useful. Sepsis-related metabolic abnormalities can cause a surge in glucose levels, which may be different from admission hyperglycemia. In a previous study, it was found that moderate glycemic control measures may benefit patients with severe hyperglycemia (>220 mg/dL) [36].

If we selected another three ranges of glucose peak values for analysis (≤140, 141–200, >200), the Kaplan-Meier survival curves were similar to our present three-group tool, but not better. We attempted to run the analysis with the peak glycemic status defined more simply, along more traditional cut-off thresholds such as those for euglycemia (<180 mg/dL) and hyperglycemia (≥180 mg/dL). The log-rank *p* value was not significant. Our data showed that U-shaped curves were noted for outcomes in 28-day mortality according to the peak glucose levels, but such result was not observed in 90-day mortality. The peak glucose level in mg/dL did not significantly correlate with mortality after risk covariable adjustment. This is possibly because the *p* value reflects a comparison of multiple groups, and therefore, often obscures important relationships, for example, the U-shaped relationship between peak glucose levels and mortality.

The mSOFA-g score can predict 90-day survival in patients with and without diabetes in the construction and validation cohort groups. In this study, the goodness of fit of the mSOFA-g score increased above 5% compared to the SOFA score in the subgroup without diabetes. This result implies that the mSOFA-g score can better predict 90-day survival in patients without diabetes. The validation data did not reveal any differences between the diabetes and non-diabetes groups, possibly because of the smaller number of cases in the validation group. A previous systemic review, which included nine studies, also pointed out that patients who suffered from hyperglycemia without diabetes had a 2.7-fold increase in hospital mortality compared to patients who suffered from hyperglycemic diabetes [37]. This was consistent with our results. Sepsis increases serum cortisol and inflammatory cytokine levels, which increase insulin resistance and gluconeogenesis [38]. Compared to diabetes-induced hyperglycemia, stress-induced hyperglycemia seems to play a more important role in inflammation severity among patients with sepsis. This may explain the increased mortality risk of patients without diabetes who suffered from hyperglycemia during the first day in ICU. Compared to patients with DM, patients without DM might have more glucose variation from the baseline to the peak glucose concentration. The complexity of glucose variation might increase the association between the peak glucose level and mortality of patients without DM. We need further studies on this issue.

The metabolic milieu and host response to sepsis are complex and the optimal blood glucose ranges are difficult to define. Hyperglycemia and diabetes have deleterious effects on neutrophil and macrophage function [39,40]. They can also affect the immune system and affect clinical outcomes [41]. Cox proportional hazards regression analysis showed that sepsis severity scores and HLA-DR expression were associated with mortality. DM can only partially account for the heterogeneity in the host response to sepsis [41]. Our results confirmed that DM alone was not associated with poor infection outcomes [42]. In our study, the DM and non-DM groups had comparable baseline immune statuses. Although admission for severe hyperglycemia was associated with increased mortality in both DM and non-DM patients in a previous report [23], we found that the phenomenon was more obvious when using the peak glucose level instead of the admission glucose level. Our data suggested that patients with P3 peak hyperglycemia had suppressed immunity, as evidenced by decreased HLA-DR expression. However, we did not include immune data in the validation cohort.

The strength of our study includes its first introduction of peak glucose levels for risk stratification in consecutive critically ill patients with sepsis. We developed a new tool for risk stratification (three-group tool) and performed detailed comparisons of the mortality hazard ratios and serial sepsis clinical scores to explore their relationships. We used both the construction and validation cohorts to confirm our findings (Appendix A). The two cohorts had different overall disease severities, comorbidity distributions, and mortalities. The results regarding the association of the peak glucose range on day 1 and the presence of diabetes mellitus with mortality were consistent. In addition, we had a subpopulation of septic patients with immune status (HLA-DR expression data), which may relate the clinical tool to possible mechanisms of immune dysfunction. Although the immune profiles were only determined in a subset of patients, this subset had comparable baseline characteristics (HbA1C, DM status, peak glucose on day 1, age, sex, BMI, and CCI) and the same inclusion/exclusion criteria compared to the total study population in the construction cohort.

There are some limitations to our study. First, the retrospective design and the limited number of cases with immune profiles preclude further analysis [22]. Although differences in serial immune status and SOFA score may be associated with differences in mortality, the mechanism of interaction between immunity and SOFA score was not within the scope of the present study. Further prospective studies with complete immune data are required to investigate this mechanism. Second, the analysis excluded 77 patients without first-day glucose measurement to avoid potential interference in our analysis. Therefore, we analyzed the differences in baseline characteristics between the exclusion and study groups (Appendix A). The exclusion group had lower APACHE II scores, peak day 1 blood glucose levels, and HbA1C levels, as well as less diabetes, chronic kidney disease, and liver disease. As the diabetes status may have been the major confounder to affect our results, we adjusted for patients with or without diabetes by stratification. In addition, we used adjusted analysis to control for other co-variables. Third, the HbA1c value (the indicator of long-term glycemic control within the previous three months) was not available for all patients. In this observational study, we enrolled all consecutive patients with sepsis, regardless of whether or not they had HbA1c data. We did not exclude patients without HbA1c reports (48.2%). However, in most institutions, HbA1c is not a routine blood test and is usually ordered based on the clinical suspicion of diabetes with poor glycemic control. In addition, for general clinical application, we did not separate sub-phenotypes of DM, such as insulin-treated DM. Yet, these DM patients may have different outcomes and require a specific approach for the management of dysglycemia [43].

## 5. Conclusions

In conclusion, our results demonstrated the relationship between simple glucose metrics and mortality, stratified by the peak glucose levels on day one in critically ill patients with sepsis. The mSOFA-g score can be used to predict 90-day mortality in patients with sepsis and can more precisely predict the 90-day mortality in patients without diabetes. The DM status and maximum glucose level on the first day of ICU admission are easy to obtain for bedside clinicians. The peak glucose level range, instead of diabetes status, can be used for mortality-risk stratification of critically ill patients with sepsis.

## Figures and Tables

**Figure 1 diagnostics-11-01798-f001:**
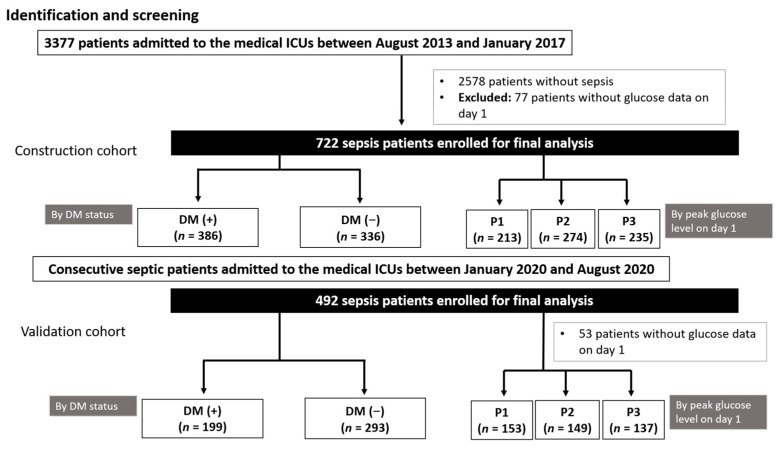
Study flowchart.

**Figure 2 diagnostics-11-01798-f002:**
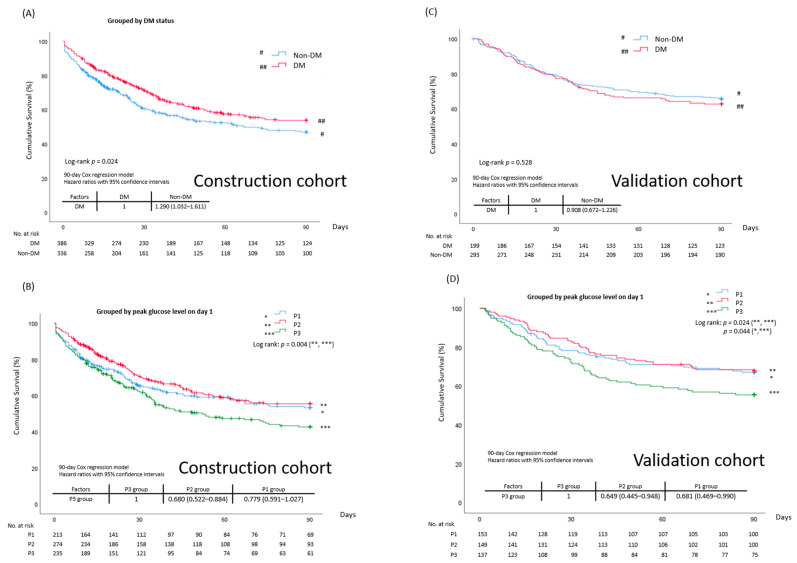
Ninety-day survival curves. (**A**) Between DM and non-DM in construction cohort; (**B**) among P1, P2, and P3 groups by peak glucose levels on day one in construction cohort; (**C**) between DM and non-DM in validation cohort; (**D**) among P1, P2, and P3 groups by peak glucose levels on day one in validation cohort. # (non-DM), ## (DM), * (P1 group), ** (P2 group), *** (P3 group). Kaplan-Meier estimates of 90-day survival according to stratification. Hazard ratios between or among groups, obtained using Cox regression models, are also shown.

**Table 1 diagnostics-11-01798-t001:** Baseline characteristics and outcomes of septic patients in construction cohort.

By DM Status or Peak Glucose Level on Day 1	DM (*n* = 386)	Non-DM (*n* = 336)	*p* ^†^	P1 Group (*n* = 213)	P2 Group (*n* = 274)	P3 Group (*n* = 235)	*p* *
Demographic characteristics, median (25th and 75th percentile)							
Age (years)	70 (61, 79)	65 (54, 78)	<0.001	66 (56,78)	68 (59, 78)	71 (60, 80)	0.024
BMI, kg/m^2^	23.7 (20.1, 26.5)	21.7 (18.7, 24.8)	<0.001	22.0 (18.2, 25.0)	22.4 (19.5, 22.5)	23.6 (20.3, 27)	<0.001
Sex (male), *n* (%)	204 (58.2)	227 (67.6)	<0.001	137 (64.3)	180 (65.7)	114 (48.5)	<0.001
APACHE II	26 (20, 32)	25 (20, 30)	0.212	24 (18.5, 30)	24 (19, 30)	27 (21, 33)	0.001
Charlson comorbidity index	2 (2, 2)	2 (1, 6)	0.521	2 (1, 5)	2 (1, 3)	2 (2, 3)	0.046
**Comorbidities, *n* (%)**							
Coronary artery disease	115 (29.8)	72 (21.4)	0.01	50 (23.5)	74 (27)	63 (26.8)	0.628
History of stroke	78 (20.2)	55 (16.4)	0.185	37 (17.4)	46 (16.8)	50 (21.3)	0.383
Hypertension	266 (69.1)	143 (42.6)	<0.001	103 (48.4)	154 (56.2)	152 (65)	0.002
COPD	55 (14.2)	46 (13.7)	0.829	21 (9.9)	42 (15.3)	38 (16.2)	0.113
Cancer	71 (18.6)	95 (28.5)	0.002	53 (25.4)	58 (21.2)	55 (23.6)	0.562
CKD	99 (37.2)	61 (25.4)	0.004	42 (26.9)	57 (30.8)	61 (37)	0.147
Liver cirrhosis	31 (8)	37 (11)	0.171	25 (11.7)	29 (10.6)	14 (6)	0.079
Diabetes mellitus	386 (100)	0 (0)	<0.001	70 (32.9)	128 (46.7)	188 (80)	<0.001
**Site of suspected infection, *n* (%)**							
Lung	237 (61.4)	221 (65.8)	0.223	130 (61)	186 (67.9)	142 (60.4)	0.151
UTI	97 (25.1)	60 (17.9)	0.018	37 (17.4)	66 (24.1)	54 (23)	0.175
Bacteremia	31 (8)	26 (7.7)	0.884	16 (7.5)	20 (7.3)	21 (8.9)	0.768
Others	80 (20.7)	66 (19.6)	0.718	130 (61)	186 (67.9)	142 (60.4)	0.151
**Baseline glucose and HbA1c, median (25th and 75th percentile)**							
HbA1c (%; mmol/mol)	7.2 (6.6, 8.4); 55 (49, 68)	5.7 (5.5, 6); 39 (37, 42)	<0.001	5.9 (5.5, 6.8); 41 (37, 51)	6.4 (5.8, 7.4); 46 (40, 57)	7.5 (6.6, 8.8); 58 (49, 73)	<0.001
Glucose (mg/dL)	219 (159, 303)	148 (121, 188)	<0.001	137 (115, 177)	152 (127, 219)	239 (151, 340)	<0.001
**Immune biomarker**							
HLA-DR expression (%) ^#^	92.4 (21.1)	91.9 (19.8)	0.699	90.4 (24)	96.6 (13.7)	88.8 (21.7)	0.013
**Mortality, *n* (%)**							
7-day mortality	40 (10.4)	58 (17.3)	0.007	31 (14.6)	26 (9.5)	41 (17.4)	0.029
28-day mortality	99 (25.6)	113 (33.6)	0.019	65 (30.5)	68 (24.8)	79 (33.6)	0.086
90-day mortality	155 (40.2)	156 (46.4)	0.09	88 (41.3)	104 (38)	119 (50.6)	0.013

Stratified by DM status or peak glucose level on day 1. ^#^ Patients with HLA-DR data on day 1: DM = 89, non-DM = 60, P1 = 38, P2 = 52, P3 = 59. ^†^ Comparison analyses between two groups by Mann-Whitney U tests or chi-squared tests for categorical variables. *p* *: Comparison analyses among three groups using one-way analysis of variance (ANOVA), with Kruskal-Wallis as a non-parametric alternative to ANOVA for non-normally distributed continuous variables or chi-square tests for categorical variables. Abbreviations: COPD, chronic obstructive pulmonary disease; CKD, chronic kidney disease; UTI, urinary tract infection; HLA-DR, human leukocyte antigen-D-related. P1 group: peak glucose level ≤ 140 mg/dL. P2 group: peak glucose level > 140 and ≤220 mg/dL. P3 group: peak glucose level > 220 mg/dL.

**Table 2 diagnostics-11-01798-t002:** Mortality comparisons between DM and non-DM groups for Cox proportional-hazards regression and logistical regression analysis in construction cohort.

			Model 1		Model 2	
DM Compared to Non-DM (Non-DM as Reference)	Crude Hazard Ratio (95% CI)	*p* Value	Adjusted Hazard Ratio (95% CI)	*p* Value	Adjusted Hazard Ratio (95% CI)	*p* Value
Cox proportional-hazards regression analysis						
7-day mortality	0.578 (0.387–0.865)	0.008	0.496 (0.324–0.758)	0.001	0.437 (0.257–0.743)	0.002
28-day mortality	0.716 (0.547–0.938)	0.015	0.679 (0.512–0.900)	0.007	0.707 (0.497–1.006)	0.054
90-day mortality	0.775 (0.621–0.969)	0.025	0.757 (0.600–0.953)	0.018	0.780 (0.589–1.034)	0.084

Model 1: Adjusted for age, sex, BMI, APACHE II, Charlson comorbidity index. Model 2: Adjusted for age, sex, BMI, APACHE II, Charlson comorbidity index, coronary artery disease, history of stroke, hypertension, COPD, cancer, CKD, liver cirrhosis.

**Table 3 diagnostics-11-01798-t003:** Cox proportional-hazards regression analysis for 90-day mortality (adjusted HR) in construction cohort.

	Model 1		Model 2	
Variable	Adjusted Hazard Ratio (95% CI)	*p* Value	Adjusted Hazard Ratio (95% CI)	*p* Value
**Total**				
Peak glucose on day 1	1.001 (1.000–1.002)	0.086	1.002 (1.001–1.003)	0.001
DM status (with compared to without)	0.757 (0.600–0.953)	0.018	0.780 (0.589–1.034)	0.084
Three-group tool (P2 compared to P3)(peak glucose level on day 1)	0.679 (0.517–0.893)	0.006	0.553 (0.393–0.777)	0.001
**With DM**				
Peak glucose on day 1	1.000 (0.998–1.002)	0.987	1.001 (0.999–1.003)	0.282
Three-group tool (P2 compared to P3)(peak glucose level on day 1)	0.722 (0.497–1.049)	0.088	0.576 (0.357–0.930)	0.024
**Without DM**				
Peak glucose on day 1	1.002 (1.001–1.003)	<0.001	1.002 (1.001–1.003)	<0.001
Three-group tool (P2 compared to P3)(peak glucose level on day 1)	0.377 (0.246–0.579)	<0.001	0.323 (0.187–0.557)	<0.001

Model 1: Adjusted for age, sex, BMI, APACHE II, Charlson comorbidity index. Model 2: Adjusted for age, sex, BMI, APACHE II, Charlson comorbidity index, coronary artery disease, history of stroke, hypertension, COPD, cancer, CKD, liver cirrhosis.

**Table 4 diagnostics-11-01798-t004:** The mSOFA-g_score and SOFA score prediction power of mortality.

Construction Group *n* = 722	* mSOFA-g Score Day 1	SOFA Score Day 1
7-day mortality β (*p*); r^2^	1.191 (<0.001); 0.115	1.206 (<0.001); 0.106
28-day mortality	1.145 (<0.001); 0.090	1.148 (<0.001); 0.077
90-day mortality	1.131 (<0.001); 0.082	1.132 (<0.001); 0.068
**Validation group *n* = 491**		
7-day mortality β (*p*); r^2^	1.133 (0.005); 0.050	1.168 (<0001); 0.062
28-day mortality	1.097 (<0.001); 0.043	1.111 (<0.001); 0.044
90-day mortality	1.172 (<0.001); 0.130	1.178 (<0.001); 0.113

* mSOFA-g score = SOFA score + FS_score (FS_score: P1 = 0; P2 = 1; P3 = 4). P1 group: peak glucose level ≤ 140 mg/dL. P2 group: peak glucose level > 140 and ≤ 220 mg/dL. P3 group: peak glucose level > 220 mg/dL.

**Table 5 diagnostics-11-01798-t005:** The mSOFA-g score and SOFA score prediction power of mortality of participants with and without diabetes.

Construction Group		
**With diabetes *n* = 386**	* mSOFA-g score day 1	SOFA score day 1
7-day mortality β (*p*); r^2^	1.133 (0.002); 0.054	1.141 (0.003); 0.047
28-day mortality	1.094 (0.002); 0.039	1.100 (0.002); 0.035
90-day mortality	1.098 (<0.001); 0.095	1.109 (<0.001); 0.047
**Without diabetes group *n* = 336**		
7-day mortality β (*p*); r^2^	1.292 (<0.001); 0.235	1.256 (<0001); 0.179
28-day mortality	1.244 (<0.001); 0.203	1.206 (<0.001); 0.141
90-day mortality	1.202 (<0.001); 0.157	1.161 (<0.001); 0.098
**Validation group**		
**With diabetes *n* = 199**	* MSOFA-g score day 1	SOFA score day 1
7-day mortality β (*p*); r^2^	1.114 (0.194); 0.028	1.081 (0.392); 0.012
28-day mortality	1.076 (0.093); 0.023	1.070 (0.154); 0.016
90-day mortality	1.203 (< 0.001); 0.150	1.192 (< 0.001); 0.117
**Validation group *n* = 293**		
7-day mortality β (*p*); r^2^	1.149 (0.008); 0.074	1.201 (0.001); 0.103
28-day mortality	1.116 (< 0.001); 0.065	1.134 (< 0.001); 0.068
90-day mortality	1.158 (< 0.001); 0.118	1.168 (< 0.001); 0.109

* mSOFA-g score = SOFA score + FS_score (FS_score: P1 = 0; P2 = 1; P3 = 4). P1 group: peak glucose level ≤ 140 mg/dL. P2 group: peak glucose level > 140 and ≤ 220 mg/dL. P3 group: peak glucose level > 220 mg/dL.

## Data Availability

The data presented in this study are available on request from the corresponding author. The data are not publicly available due to patients’ privacy.

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
