# Peer review of "Application of Peak Glucose Range and Diabetes Status in Mortality Risk Stratification in Critically Ill Patients with Sepsis"

_diagnostics, 2021, doi:10.3390/diagnostics11101798_

Round 1

Reviewer 1 Report

The study is accepted after minor revision. 
The research design is appropriate and the results are clear. However, the purpose of the research is not stated clearly in the article.

Author Response

Reviewer#1

Comments and Suggestions for Authors

The study is accepted after minor revision.

The research design is appropriate and the results are clear. However, the purpose of the research is not stated clearly in the article.

A: Many thanks for your valuable comment and encouragement. 

We have revised the Introduction paragraph to add the purpose of the research (page 2 , paragraph 2 )

Sepsis can induce stress hyperglycemia in both patients with and without DM [16]. Patients with sepsis who are administered intensive insulin treatment also have a high risk of hypoglycemia and hyperglycemia[17]. Peak glucose values may be associated with sepsis outcomes, but are seldom discussed. With sepsis progression or resolution, the peak glucose level on the first day of ICU admission may differ. We hypothesized that there is an association between peak glucose levels and mortality risk in critically ill septic patients with or without DM. Their association with outcomes could be related to the clinical parameters or immune status. The purpose of our study is to investigate the application of peak glucose range and diabetes status in mortality risk stratification in critically ill patients with sepsis. In addition, we proposed a new tool, the modified Sequential Organ Failure Assessment-glucose (mSOFA-g) score, to facilitate mortality risk stratification.  

Reviewer 2 Report

Hung and Feng et al. drafted the manuscript entitled ”Application of Peak Glucose Range and Diabetes Status in 2 Mortality Risk Stratification in Critically Ill Patients with Sepsis” for the original article. The study group used the peak glucose level as a predictor for the 90 day mortality in septic patients independent of the DM status of the subjects. The modified SOFA-g score, initiated from the study group, provided a better predicting power for 90 days mortality in the medical center from the cohort study. The result provided the importance of peak glucose and the mortality in critical ill patients. There are only several issues needed to be addressed:

  1. Introduction
  2. Materials and methods:
    a. Please define the definition of the mortality.
    b. Please list the definition of the comorbidities such as CAD, COPD, CHF in the definition section (from the medical records.. etc).
  3. Please add the reference of the P1, P2, P3 or the reason why the study group set the cut-off value (refence 33, 34).
  4. Please document that the study group record the severity score on day 1 and day 3, because the table in the supplement document the serial change.

  1. Results:
  2. Table 1 did not demonstrate the percentage of HLA-DR in different subgroups, which were demonstrated in the table S6. Please revise the table 1 accordingly.
  3. In the table 4, the authors illustrated the m-SOFA-g score. In line 227, the score for peak glucose level was 1,2,4 in P1,P2 and P3. What is the reason? Is there any regression executed by the study group or is there a reference? Please address the issue.
    d. The quality of figure 2 is below the academic standard. Please Re-send the figure according to the MDPI guideline.

    4. Discussion:
    a. Line 298: Is there misuse for U-shape? From the authors result, the 90-day mortality was the highest in P3 group in both construction and validation cohort. Please verify the term.
  4. The authors need to discuss the association between the DM and the peak glucose level in the cohort.
    1)From the cohort study, the patients with DM status had better survival. Is there possible explanation? Such as bias from single institute?
    2) In comparison with the non-DM subjects, the glucose variation from the base line to the peak glucose concentration might be less in DM subjects. Is the glucose complexity in non-DM subjects more associated the mortality?
